REGISTERED REPORT PROTOCOL

# Core warming of coronavirus disease 2019 (COVID-19) patients undergoing mechanical ventilation—A protocol for a randomized controlled pilot study

Nathaniel Bonfanti[1,2], Emily Gundert[1,2], Anne M. Drewry[3], Kristina Goff[4], Roger Bedimo[5,6], Erik Kulstad[1] *

1 Department of Emergency Medicine, University of Texas, Southwestern Medical Center, Dallas, TX, United States of America, 2 Department of Anesthesia/Critical Care, University of Texas, Southwestern Medical Center, Dallas, TX, United States of America, 3 Department of Anesthesiology, Washington University School of Medicine, St. Louis, MO, United States of America, 4 Department of Anesthesiology and Pain Management, University of Texas, Southwestern Medical Center, Dallas, TX, United States of America, 5 Infectious Diseases Section, VA North Texas Health Care System, Dallas, TX, United States of America, 6 Department of Internal Medicine, University of Texas, Southwestern Medical Center, Dallas, TX, United States of America

* erik.kulstad@utsouthwestern.edu

## Abstract

### Background

Coronavirus disease 2019 (COVID-19), caused by the virus SARS-CoV-2, is spreading rapidly across the globe, with little proven effective therapy. Fever is seen in most cases of COVID-19, at least at the initial stages of illness. Although fever is typically treated (with antipyretics or directly with ice or other mechanical means), increasing data suggest that fever is a protective adaptive response that facilitates recovery from infectious illness.

### Objective

To describe a randomized controlled pilot study of core warming patients with COVID-19 undergoing mechanical ventilation.

### Methods

This prospective single-site randomized controlled pilot study will enroll 20 patients undergoing mechanical ventilation for respiratory failure due to COVID-19. Patients will be randomized 1:1 to standard-of-care or to receive core warming via an esophageal heat exchanger commonly utilized in critical care and surgical patients. The primary outcome is patient viral load measured by lower respiratory tract sample. Secondary outcomes include severity of acute respiratory distress syndrome (as measured by PaO2/FiO2 ratio) 24, 48, and 72 hours after initiation of treatment, hospital and intensive care unit length of stay, duration of mechanical ventilation, and 30-day mortality.

**Data Availability Statement:** All relevant data from this study will be made available upon study completion.

**Funding:** EK declares equity interest in Attune Medical. The funders had no role in study design, data collection and analysis, decision to publish, or preparation of the manuscript.

**Competing interests:** EK declares equity interest in Attune Medical. This does not alter our adherence to PLoS ONE policies on sharing data and materials.

## Results

Resulting data will provide effect size estimates to guide a definitive multi-center randomized clinical trial. ClinicalTrials.gov registration number: NCT04426344.

## Conclusions

With growing data to support clinical benefits of elevated temperature in infectious illness, this study will provide data to guide further understanding of the role of active temperature management in COVID-19 treatment and provide effect size estimates to power larger studies.

## Introduction

Traditionally, fever has been treated because its metabolic costs were felt to outweigh its potential physiologic benefit in an already stressed host [1]. However, increasing data suggest that fever may be a protective adaptive response that should be allowed to run its course under most circumstances [2,3]. Higher early fever is associated with a lower risk of death among patients with an ICU admission diagnosis of infection [4,5]. Fever may enhance immune-cell function [6,7], inhibit pathogen growth [8–10], and increase the activity of antimicrobial drugs [11]. Fever potentially benefits infected patients via multiple mechanisms; *in vitro* and animal studies have shown that elevated temperatures augment immune function, increase production of protective heat shock proteins, directly inhibit microorganism growth, reduce viral replication, and enhance antibiotic effectiveness [3,12]. More rapid recoveries are observed from chickenpox [13], malaria [14], and rhinovirus [15] infections with avoidance of antipyretic medication, and many innate and adaptive immunological processes are accelerated by fever [16–18].

Randomized controlled trials have consistently failed to find benefits to treating fever of infectious etiology [16,19–25]. Reducing patient temperature to below normal in sepsis likewise has been found to be of no benefit, or harmful [26,27]. On the other hand, warming appears to have substantial benefits in sepsis. Multiple aspects of both humoral and cellular immunity (including antibody production, T lymphocyte trafficking, T cell adhesion and migration, heat shock protein 90 (Hsp90)-induced α4 integrin activation and signaling, and macrophage function) are boosted by elevated temperature [28]. A retrospective cohort study evaluating 1,264 patients requiring mechanical ventilation found that high fever ($\geq39.5˚C$) was associated with increased risk for mortality in mechanically ventilated patients; however, in patients with sepsis, moderate fever ($38.3˚C$-$39.4˚C$) was protective, and antipyretic medication was not associated with changes in outcome [29]. Prospective data show that afebrile patients have higher 28-day mortality (37.5% vs 18.2%), increased acquisition of secondary infections (35.4% vs. 15.9%), and suppressed HLA-DR expression suggestive of monocyte dysfunction over time [30]. As recently as the 1910's, the "malaria fever cure" (inducing fever to treat a range of conditions, an approach known as "pyrotherapy") was widespread, with the originator of the idea receiving the Nobel Prize in Medicine or Physiology in 1927 [31,32]. Currently, the UK National Institute for Health and Care Excellence (NICE) recommend not using antipyretic agents "with the sole aim of reducing body temperature in children with fever [16,33]". Actively inducing hyperthermia by directly heating the body has been used in cancer treatment, with minimal adverse effects [34–37]. Hyperthermia has been found to have positive impacts on the immune system, causing increased levels of heat-shock proteins [28,38,39], which are directly related to antigen presentation and cross-presentation, activation of macrophages and lymphocytes, and activation and maturation of dendritic cells [40]. A

pilot study of external warming of septic patients (ClinicalTrials.gov Identifier: NCT02706275) has recently been completed.

Many viruses replicate more robustly at cooler temperatures, such as those found in the nasal cavity (33–35˚C) than at warmer core body temperature (37˚C) [41–45]. Coronavirus disease 2019 (COVID-19) currently has limited treatment options besides dexamethasone [46], but its causative virus (SARS-CoV-2) may behave similarly to other viruses susceptible to temperature changes [47]. Simulations of the receptor binding domain (RBD) of SARS-CoV-2 found high flexibility near the binding site, suggesting that the RBD will have a high entropy penalty upon binding angiotensin-converting enzyme II (ACE2), and that consequently, the virus may be more temperature-sensitive in terms of human infection than other coronaviruses [48]. Notably, fever has often abated by the time a COVID-19 patient requires mechanical ventilation [49]. Additionally, patients with severe COVID-19 tend to have a high viral load and a long virus-shedding period, suggesting that the viral load of SARS-CoV-2 might be a useful marker for assessing disease severity and prognosis [50]. The aim of this study is to determine the effect of active core warming patients diagnosed with COVID-19 and undergoing mechanical ventilation. We hypothesize that active core warming will reduce the severity of acute respiratory distress syndrome, reduce the duration of mechanical ventilation, and improve survival compared to standard of care.

## Study objectives

The purpose of the proposed pilot study is to determine if core warming improves respiratory physiology of mechanically ventilated patients with COVID-19, allowing earlier weaning from ventilation, and greater overall survival.

### Primary objective

1. Determine the change in viral load measured in lower respiratory tract sample after implementation of core warming of ventilated patients, and compare this change to patients undergoing standard care.

### Secondary objectives

1. Measure the impact of esophageal core warming on severity of acute respiratory distress syndrome as measured by PaO2/FiO2 ratio 24, 48, and 72 hours after initiation, and compare this to standard care.

2. Compare the duration of mechanical ventilation of patients treated with core warming to patients treated with standard care.

3. Compare the length of ICU and hospital stay of patients treated with core warming to patients treated with standard care.

4. Compare the 30-day mortality of patients treated with core warming to patients treated with standard care.

## Methods

This is a single-center pilot study to evaluate if core warming improves respiratory physiology of mechanically ventilated patients with COVID-19, allowing earlier weaning from ventilation, and greater overall survival. The protocol was reviewed and approved by the Institutional

Review Board of Washington University. The study is listed on ClinicalTrials.gov with identifier NCT04426344. This prospective, randomized study will include 20 patients diagnosed with COVID-19, and undergoing mechanical ventilation for the treatment of respiratory failure. Patients will be randomized in a 1:1 fashion with 10 patients (Group A) randomized to undergo core warming with an esophageal heat transfer device, and the other 10 patients (Group B) serving as the control group. Patients randomized to Group A will have the esophageal heat transfer device placed in the ICU or other clinical environment in which they are being treated after enrollment and provision of informed consent from appropriate surrogate or legally authorized representative. This study is posted on ClinicalTrials.gov with registration number: NCT04426344. The IRB of Washington University, St. Louis, is performing full review of the final protocol and expected to provide approval; the study will not start prior to IRB approval.

## Screening

Subjects will be recruited from the ICU or other clinical environment in which they are being treated (Emergency Department, step-down unit, etc.). Patients will be identified by the PI or other study investigators/coordinators as available, and will be restricted to those who have been undergoing mechanical ventilation for three days or less. All patients without a DNR order with a diagnosis of COVID-19 and meeting inclusion criteria will be eligible for screening for any exclusion criteria. Written informed consent for the research study will be obtained from patient's surrogate or legally authorized representative prior to enrollment. A formal screening log will be maintained for the trial, and available data on patients not entered into the study will be compared to those entered into the study. Baseline variables of patients entered into the study will additionally be compared by randomization arm.

## Study intervention and monitoring

Participants who have a signed research study consent form (via surrogate or legally authorized representative) will be randomized in a 1:1 fashion to core warming or to standard of care (standard temperature management and treatment). The esophageal heat transfer device will be used according to FDA 510(k) labeling (for patient warming). Patient temperature measurements will be collected for both the device and standard-of-care arms during the study period (up to 72 hours). Device placement will be performed using standard protocol per instructions for use. The esophageal heat transfer device will be set to 42˚C temperature after initial placement, and maintained at 42˚C for the duration of treatment. All patients will have usual standard of care labs, vital signs, and imaging for patients in critical condition undergoing mechanical ventilation in the ICU. Specific parameters to be measured include PaO2 at regular intervals appropriate for patients undergoing mechanical ventilation, and FiO2 at the time of obtaining blood gases for PaO2 measurement, to allow calculation of P/F ratio.

Control group patients will be managed as per standard of care currently utilized in the ICU, which will include the use of other methods of temperature management as warranted. This would include warming with a forced air blanket only in hypothermic patients (core temperature < 36˚C) or antipyretic therapy for febrile patients, as requested by the treating physician. Episodes of hypothermia are infrequent and transient in this population, and the current standard of care generally utilizes a permissive approach to fever (allowing patients to remain mildly febrile) which will continue in the control group without modification (no intentional elevation of temperature will be provided in the control group).

## Study endpoints

The purpose of this pilot study is to determine initial estimates on outcomes (viral load, PaO2/FiO2 ratio, duration of mechanical ventilation, and mortality) in order to determine adequate sample size to properly power definitive studies. Measurements will be compared at time points 24, 48, and 72 hours after initiation. Sampling for viral measurements will utilize lower respiratory tract samples, as these have been shown to be of greater sensitivity and reliability for patient monitoring [47,51,52].

## Primary study endpoints

*The primary endpoint of this study will be*:

1. Viral load measured in lower respiratory tract sample 72 hours after initiation of core warming

   *Secondary study endpoints include*:

1. PaO2/FiO2 ratio 24, 48, and 72 hours after initiation of core warming

2. Duration of mechanical ventilation

3. Duration of ICU and hospital stay

4. Patient mortality

## Inclusion criteria

1. Patients above the age of 18 years old.

2. Patients with a diagnosis of COVID-19 on mechanical ventilation.

3. Patient maximum baseline temperature (within previous 12 hours) < 38.3˚C.

4. Patients must have a surrogate or legally authorized representative able to understand and critically review the informed consent form.

## Exclusion criteria

1. Patients with contraindication to core warming using an esophageal core warming device.

2. Patients known to be pregnant.

3. Patients with <40 kg of body mass.

4. Patients with DNR status.

5. Patients with acute stroke, post-cardiac arrest, or multiple sclerosis.

## Subject recruitment

Subjects will be recruited from the ICU or other clinical environment in which they are being treated (Emergency Department, step-down unit, etc.). Patients will be identified by the PI or other study investigators/coordinators as available. All patients without a DNR order with a diagnosis of COVID-19 and meeting inclusion criteria will be eligible for screening for any exclusion criteria. Written informed consent for the research study will be obtained from patient's surrogate or legally authorized representative prior to enrollment. If a patient enrolled

in the study gains the capacity to consent for him/herself while the study is in progress, the patient will be approached by a study team member and the consent document will be presented directly to the patient. All questions the patient might have will be answered. The patient will be given the opportunity to either withdraw from the study or sign the consent form. The patient will be informed that his or her decision to withdraw from the study will not affect his or her medical care

## Duration of study participation

Participants will be involved for approximately 1 month, including screening, treatment, and follow-up. After consent, patient participation in the intervention phase will last 72 hours for active treatment. The follow up for determination of outcome and duration of mechanical ventilation will occur at 1-month post-treatment. Additional data will be collected via chart review.

## Total number of subjects and sites

This single-site study aims to recruit and randomize 20 patients. It is expected that up to 30 subjects may be consented in order to produce 20 randomized & evaluable subjects.

## Core temperature modulation

Core temperature control and warming will be performed with a commercially available esophageal heat exchange device (ensoETM, Attune Medical, Chicago, IL). This device is currently used world-wide for various patient temperature management goals, including post-cardiac arrest therapeutic hypothermia [53–56], warming of burn patients [57], warming general surgical patients [58], cooling traumatic brain injury [59], cooling heat stroke [60], and the treatment of central fever [61,62]. The device is a multi-chambered silicone tube placed in the esophagus and connected to a heat exchanger to provide heat transfer to or from a patient (video available at https://vimeo.com/306506411). Modulation and control of the patient's temperature is achieved by adjusting setpoint on the external heat exchanger, which in turn controls the circulating water temperature. Two lumens of the device connect to the external heat exchanger, while a third central lumen provides stomach access for gastric decompression or tube feeding. It is a single-use, disposable, non-implantable device with an intended duration of use of 72 hours or less.

## Intervention regimen

Patients who are randomized to core warming will have the esophageal heat transfer device placed in the ICU or other treatment area where patient is undergoing mechanical ventilation. The device will remain in place until the study is completed (72 hours). The device will be set to 42˚C for the duration of the study period. It is expected that patient temperature will increase from baseline by 1˚C to 2˚C, but due to ongoing heat loss from the patient, the expected maximum patient temperature is below 39˚C. The time course of illness of COVID-19 is such that patients often no longer have fever by the time of mechanical ventilation [41]. If patient temperature increases above this range and reaches 40˚C, the device will be set to an operating temperature of 40˚C, thereby preventing any further increase in patient temperature. Patient temperature will be followed at intervals per standard of care in the intensive-care setting for mechanically ventilated patients (typically hourly).

## Blinding

Due to the nature of this study, the physicians will not be blinded to the randomization assignment, however participants will be blinded. Once a subject is randomized, the research team will receive the randomization assignment (core warming or standard of care) and proceed with the procedures per the assignment.

## Data collection

- Demographics (including sex/gender, race, ethnicity, and age via date of birth)

- Past medical history, social history, physical exam findings and physicians notes

- Concurrent medications

- Physical exam

- Vital signs: temperature, blood pressure, heart rate, respiration rate, height and weight

- Clinical labs: complete blood count (CBC), chemistry profiles, liver function tests, inflammatory markers (CRP, ferritin), d-dimer, arterial blood gas for determination of PaO2

- Upper (nasopharyngeal) and lower (tracheal aspirate, sputum) respiratory tract viral load (cycle threshold)

- Severity of illness: APACHE III, sequential organ failure assessment (SOFA) scoring systems

- Ventilator settings

- Pregnancy test for women of childbearing age

- Adverse events or unanticipated problems

Data will be collected via chart review, and is expected to be available from routinely obtained laboratory and vital sign data recorded at routine intervals (i.e., when labs are drawn for routine care in the ICU).

## Schedule of procedures and data collection

| Study Phase | Screening | Randomization/Intervention Phase | | | Follow- up |
|---|---|---|---|---|---|
| Study Days | Day -1 to 0 | Day 0 | Day 1–2 | Day 7, 14 | Day 30 |
| Informed Consent | X | | | | |
| Review Inclusion/Exclusion Criteria | X | | | | |
| Demographics | X | | | | |
| Medical History/Interim History* | X | | X | | X |
| Physical Examination* | X | | X | | X |
| Vital Signs: Temperature, BP, HR, RR* | X | | X | | X |
| Height and Weight | X | | | | |
| Pregnancy Test | X | | | | |
| Clinical Laboratory Evaluation | X | X | X | | |
| Respiratory tract viral load | | | | X | |
| Ventilator settings | X | X | X | | |
| APACHE III and SOFA scores | X | X | X | | |

(*Continued*)

| Study Phase | Screening | Randomization/Intervention Phase | | | Follow- up |
|---|---|---|---|---|---|
| Study Days | Day -1 to 0 | Day 0 | Day 1–2 | Day 7, 14 | Day 30 |
| Clinical Imaging | X | X | X | | |
| Prior/Concomitant Medications | X | | | | X |
| Randomization | | X | | | |
| Temperature monitoring | | X | X | | |
| PaO2, FiO2, parameter recording | | X | X | | |
| Discharge | | | X | | |
| Adverse Event / Unanticipated Problems Assessment | | X | X | | X |

* Interim medical history, physical exam, and vitals will be collected via chart review from routine clinical care.

## Sample size and power determination

Based on a prior study in patients with sepsis, a maximum temperature of 38.3˚C to 39.4˚C was associated with survival (aHR 0.61 [95% CI, 0.39–0.99]) [29]. However, the effect of warming specific to COVID-19 patients remains uncertain, and as such, it is not possible to accurately perform a power calculation for this pilot study. It is believed that a total of 10 patients for each group will yield the sufficient pilot data to make an appropriate conclusion regarding the potential utility of core warming in reducing viral load, improving pulmonary physiology, reducing mechanical ventilation duration, and increasing patient survival. It is anticipated that data from this pilot study can be used for planning future larger studies.

## Statistical methods

We will utilize standard measures to report outcomes and measure differences between groups. Specifically, we will use descriptive statistics, including mean (standard deviation) and median (interquartile range). Kaplan-Meier plots of important time to event outcomes and measures will be produced. Normality will be assessed using histograms and the Kolmogorov–Smirnov test. Formal hypothesis testing is not planned for this pilot feasibility study.

## Efficacy analysis

This is a pilot feasibility study to determine the potential role of core warming during COVID-19 treatment.

## Interim safety analysis

All subjects entered into the study and randomized at the baseline timepoint will have detailed information collected on adverse events for the overall study safety analysis. An interim safety analysis will be performed after the first 10 subjects are enrolled in the trial. At this time the safety and tolerability of the study device will be assessed and if deemed safe and appropriate, enrollment will continue to 20 subjects.

## Subject population for analysis

All patients enrolled, randomized to a study arm, and completed in the study will be included for analysis.

## Conclusion

We describe, before the initiation of any data collection, our approach to obtaining and analyzing data from a pilot randomized-controlled trial of core warming patients undergoing mechanical ventilation due to COVID-19. We anticipate this framework will enhance the utility of the reported results and provide a solid basis from which to design and execute subsequent investigations.

## Supporting information

**S1 File. Case report form—core warming COVID-19.**
(DOCX)

**S2 File. Consent template.**
(DOCX)

**S3 File. Safety and monitoring—protocol—core warming in COVID-19.**
(DOCX)

**S4 File. SPIRIT-checklist-core warming COVID.**
(DOC)

## Author Contributions

**Conceptualization:** Nathaniel Bonfanti, Emily Gundert, Anne M. Drewry, Roger Bedimo, Erik Kulstad.

**Investigation:** Anne M. Drewry.

**Methodology:** Nathaniel Bonfanti, Anne M. Drewry, Kristina Goff, Roger Bedimo.

**Project administration:** Erik Kulstad.

**Supervision:** Anne M. Drewry.

**Writing – original draft:** Nathaniel Bonfanti, Roger Bedimo, Erik Kulstad.

**Writing – review & editing:** Nathaniel Bonfanti, Emily Gundert, Anne M. Drewry, Kristina Goff, Roger Bedimo, Erik Kulstad.

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
