## [Decision Letter · Decision Letter 0]

9 Sep 2020

PONE-D-20-12959

Core warming of coronavirus disease 2019 (COVID-19) patients undergoing mechanical ventilation: protocol for a randomized controlled pilot study

PLOS ONE

Dear Dr. Kulstad,

Thank you for submitting your manuscript to PLOS ONE. After careful consideration, we feel that it has merit but does not fully meet PLOS ONE’s publication criteria as it currently stands. Therefore, we invite you to submit a revised version of the manuscript that addresses the points raised during the review process.

Specifically, both reviewers raised overlapping concerns about the proposed study design and the statistical methodology. 

We look forward to receiving your revised manuscript.

Kind regards,

Richard Hodge

Associate Editor

PLOS ONE

Journal Requirements:

2. At this time, we ask that you please specifically state that the protocol was reviewed and approved by the Institutional Review Board of Washington University

"EK declares equity interest in Attune Medical."           

"I have read the journal's policy and the authors of this manuscript have the following

competing interests: EK declares equity interest in Attune Medical."

Reviewers' comments:

Reviewer's Responses to Questions

**Comments to the Author**

1. Does the manuscript provide a valid rationale for the proposed study, with clearly identified and justified research questions?

Reviewer #1: Yes

Reviewer #2: No

2. Is the protocol technically sound and planned in a manner that will lead to a meaningful outcome and allow testing the stated hypotheses?

Reviewer #1: Yes

Reviewer #2: No

3. Is the methodology feasible and described in sufficient detail to allow the work to be replicable?

Reviewer #1: Yes

Reviewer #2: No

4. Have the authors described where all data underlying the findings will be made available when the study is complete?

Reviewer #1: Yes

Reviewer #2: No

5. Is the manuscript presented in an intelligible fashion and written in standard English?

Reviewer #1: Yes

Reviewer #2: Yes

6. Review Comments to the Author

You may also provide optional suggestions and comments to authors that they might find helpful in planning their study.

Reviewer #1: This study looks to carry out a small pilot project of 20 severely ill and ventilated COVID-19 patients equally randomised into one of two arms a) raising body temperature to an elevated degree but below 40C b) treatment as usual. The study aims to generate data regarding important clincial parameters in order to design a definitive clinical trial at a later date. They have already registered the trial at https://clinicaltrials.gov/ct2/show/NCT04426344.

I think the analysis would be enhanced if the authors state that they will produce Kaplan-Meier plots of important time to event outcomes/measures eg death, time on ventilation etc. Further I would like to see mention of a sceening log for this trial and a sentence of how they will report on this in order to give a fuller picture of what they will do/have done. Furthermore given the need for this to be an unblinded study, being able to compare who did and did not get entered into the study and to which arm, if entered, would be useful. I'd also like to see the 1st Exclusion removed as this is simply a negation of the last Inlcusion criteria, furthermore it doesnt exist on the kindly provided CRF.

They may also wish to slightly amend their introduction about there being no treatment in light of the RECOVERY trial result https://www.medrxiv.org/content/10.1101/2020.06.22.20137273v1 and https://www.nejm.org/doi/10.1056/NEJMoa2021436 (full disclosure my brother is a co-author on these reports)

Reviewer #2: The authors outline a protocol for a randomized controlled pilot study of core temperature warming in mechanically ventilated patients with COVID-19 and its effects on viral load in the endotracheal tube (primary outcome) and several additional clinical outcomes including disease severity, hospital and intensive care unit length of stay, duration of

mechanical ventilation, and 30-day mortality. Though the rationale for the study is plausible, and certainly should be explored further, the manuscript does not provide adequate information on several key areas outlined below. Please note, it is very difficult to make comments on the manuscript without page numbers or line numbers.

1. Why was the viral load in the endotracheal tube chosen as the primary outcome? There is data on nasal/NP viral kinetics, but very little in endotracheal samples, and there is a high probability that this measurement will not demonstrate an appreciable difference between groups.

2. There is no description of the sample size calculation relative to the outcomes. Given the small sample size proposed, a more realistic outcome would be feasibility.

3. Duration of the study seems way too short to assess the outcomes being investigated – especially hospitalization duration and mortality.

4. There is no consideration for the duration of mechanical ventilation at the time of enrollment – there will be a meaningful difference in ability to assess outcomes depending on time since illness onset and time since ventilation.

7. PLOS authors have the option to publish the peer review history of their article (what does this mean?). If published, this will include your full peer review and any attached files.

Reviewer #1: **Yes: **Greg Fegan

Reviewer #2: No

---

## [Author Response · Author response to Decision Letter 0]

15 Sep 2020

Thank you for the opportunity to submit our revised manuscript, PONE-D-20-12959,

Core warming of coronavirus disease 2019 (COVID-19) patients undergoing mechanical ventilation: protocol for a randomized controlled pilot study.

Journal Requirements:

We believe we now meet your style requirements. 

2. At this time, we ask that you please specifically state that the protocol was reviewed and approved by the Institutional Review Board of Washington University

We have added this to the manuscript under the Methods section.

We are happy to provide our data, however it obviously will not be available for quite some time, since the study still has to be performed and completed. We hope that this protocol paper can nevertheless be published prior to completion of the study and analysis of the data. 

"EK declares equity interest in Attune Medical." 

This amended role of funder statement is now included.

"I have read the journal's policy and the authors of this manuscript have the following

competing interests: EK declares equity interest in Attune Medical."

 Updated competing interests statement is now included.

These have been added. 

Reviewers' comments:

Reviewer's Responses to Questions

Comments to the Author

1. Does the manuscript provide a valid rationale for the proposed study, with clearly identified and justified research questions?

Reviewer #1: Yes

Reviewer #2: No

2. Is the protocol technically sound and planned in a manner that will lead to a meaningful outcome and allow testing the stated hypotheses?

Reviewer #1: Yes

Reviewer #2: No

3. Is the methodology feasible and described in sufficient detail to allow the work to be replicable?

Reviewer #1: Yes

Reviewer #2: No

4. Have the authors described where all data underlying the findings will be made available when the study is complete?

Reviewer #1: Yes

Reviewer #2: No

5. Is the manuscript presented in an intelligible fashion and written in standard English?

Reviewer #1: Yes

Reviewer #2: Yes

6. Review Comments to the Author

You may also provide optional suggestions and comments to authors that they might find helpful in planning their study.

Reviewer #1: This study looks to carry out a small pilot project of 20 severely ill and ventilated COVID-19 patients equally randomised into one of two arms a) raising body temperature to an elevated degree but below 40C b) treatment as usual. The study aims to generate data regarding important clincial parameters in order to design a definitive clinical trial at a later date. They have already registered the trial at https://clinicaltrials.gov/ct2/show/NCT04426344.

I think the analysis would be enhanced if the authors state that they will produce Kaplan-Meier plots of important time to event outcomes/measures eg death, time on ventilation etc. Further I would like to see mention of a sceening log for this trial and a sentence of how they will report on this in order to give a fuller picture of what they will do/have done. Furthermore given the need for this to be an unblinded study, being able to compare who did and did not get entered into the study and to which arm, if entered, would be useful. I'd also like to see the 1st Exclusion removed as this is simply a negation of the last Inlcusion criteria, furthermore it doesnt exist on the kindly provided CRF.

They may also wish to slightly amend their introduction about there being no treatment in light of the RECOVERY trial result https://www.medrxiv.org/content/10.1101/2020.06.22.20137273v1 and https://www.nejm.org/doi/10.1056/NEJMoa2021436 (full disclosure my brother is a co-author on these reports)

Thank you for your review and resulting suggestions for improvement. 

We have specified that Kaplan-Meier plots will be produced.

We have now included the use of a formal screening log, and our plans to report on screening results, as well as entry versus non-entry into the clinical study.

We have removed the first exclusion criteria as suggested.

We have now amended the abstract, and modified the introduction to include the mention of this treatment option. 

Reviewer #2: The authors outline a protocol for a randomized controlled pilot study of core temperature warming in mechanically ventilated patients with COVID-19 and its effects on viral load in the endotracheal tube (primary outcome) and several additional clinical outcomes including disease severity, hospital and intensive care unit length of stay, duration of mechanical ventilation, and 30-day mortality. Though the rationale for the study is plausible, and certainly should be explored further, the manuscript does not provide adequate information on several key areas outlined below. Please note, it is very difficult to make comments on the manuscript without page numbers or line numbers.

Thank you for your review and suggestions for improvement. We have now included page numbers and line numbers.

1. Why was the viral load in the endotracheal tube chosen as the primary outcome? There is data on nasal/NP viral kinetics, but very little in endotracheal samples, and there is a high probability that this measurement will not demonstrate an appreciable difference between groups.

As further clarification, we do not plan samples from the endotracheal tube itself, but rather lower respiratory tract samples, which have been shown to be more reliable and sensitive than upper airway samples. We have provided further references on this topic in the manuscript.

2. There is no description of the sample size calculation relative to the outcomes. Given the small sample size proposed, a more realistic outcome would be feasibility.

We do not intend to power this pilot for superiority. We have further emphasized the feasibility aspect of this study.

3. Duration of the study seems way too short to assess the outcomes being investigated – especially hospitalization duration and mortality.

We have utilized standard intensive care measures, where ICU, hospital, and 28 or 30 day mortality is most common. Because most severity of illness scoring systems are based on either in-hospital or 30 day mortality, we believe that deviating from this standard may introduce further interpretability challenges.

4. There is no consideration for the duration of mechanical ventilation at the time of enrollment – there will be a meaningful difference in ability to assess outcomes depending on time since illness onset and time since ventilation.

Yes, we agree. We expect that randomizatioin may help balance differences in duration of mechanical ventilation at the time of enrollment; however, we have now specified that patients will be screened only if undergoing mechanical ventilation for three days or less. 

7. PLOS authors have the option to publish the peer review history of their article (what does this mean?). If published, this will include your full peer review and any attached files.

Do you want your identity to be public for this peer review? For information about this choice, including consent withdrawal, please see our Privacy Policy.

Reviewer #1: Yes: Greg Fegan

Reviewer #2: No

---

## [Decision Letter · Decision Letter 1]

26 Oct 2020

PONE-D-20-12959R1

Core warming of coronavirus disease 2019 (COVID-19) patients undergoing mechanical ventilation: protocol for a randomized controlled pilot study

PLOS ONE

Dear Dr. Kulstad,

Thank you for submitting your manuscript to PLOS ONE. After careful consideration, we feel that it has merit but does not fully meet PLOS ONE’s publication criteria as it currently stands. Therefore, we invite you to submit a revised version of the manuscript that addresses the points raised during the review process.

We look forward to receiving your revised manuscript.

Kind regards,

Steven Eric Wolf, MD

Academic Editor

PLOS ONE

Additional Editor Comments (if provided):

Editor - Thank you for resubmitting your paper. As promised, I sent it back to the original referees who are now almost completely satisfied save a few minor issues. Please carefully consider the comments below and reply directly to each in a cover letter with appropriate marked and linked changes to the manuscript. I look forward to receiving the next version which I will handle personally for timeliness.

Reviewers' comments:

Reviewer's Responses to Questions

**Comments to the Author**

1. Does the manuscript provide a valid rationale for the proposed study, with clearly identified and justified research questions?

Reviewer #1: Yes

Reviewer #2: Partly

2. Is the protocol technically sound and planned in a manner that will lead to a meaningful outcome and allow testing the stated hypotheses?

Reviewer #1: Yes

Reviewer #2: Partly

3. Is the methodology feasible and described in sufficient detail to allow the work to be replicable?

Reviewer #1: Yes

Reviewer #2: Yes

4. Have the authors described where all data underlying the findings will be made available when the study is complete?

Reviewer #1: Yes

Reviewer #2: Yes

5. Is the manuscript presented in an intelligible fashion and written in standard English?

Reviewer #1: Yes

Reviewer #2: Yes

6. Review Comments to the Author

You may also provide optional suggestions and comments to authors that they might find helpful in planning their study.

Reviewer #1: The authors have satisfactorily accommodated all my previous request. This is the 2nd time I have seen this paper and I am happy with the paper to go ahead.

Reviewer #2: Thank you for your response. You have addressed my comments but there remain a few points that require additional thought/consideration:

1. I am still concerned about the primary endpoint – will any lower respiratory tract specimen be used? There are data suggesting differences between BAL, endotracheal tube aspirates, and sputum. What is a patient is no longer intubated at 72 hours? Will their BAL/endotracheal aspirate be compared to a sputum sample at 72 hrs? Sample type appears to be a confounder for interpretation of the results. Additionally, in line 159 the endpoint is still listed as endotracheal aspirate.

2. There is no discussion of how administration of antiviral therapy may affect the virologic endpoint. The population of interest (mechanically ventilated adults) would not uniformly receive remdesivir given current guidelines, however, some may either before study enrollment or after. Stratification could be an option.

7. PLOS authors have the option to publish the peer review history of their article (what does this mean?). If published, this will include your full peer review and any attached files.

Reviewer #1: **Yes: **Greg Fegan

Reviewer #2: No

---

## [Author Response · Author response to Decision Letter 1]

26 Oct 2020

Thank you for the opportunity to submit our revised manuscript, PONE-D-20-12959,

Core warming of coronavirus disease 2019 (COVID-19) patients undergoing mechanical ventilation: protocol for a randomized controlled pilot study.

Reviewer #2: Thank you for your response. You have addressed my comments but there remain a few points that require additional thought/consideration:

Thank you for your review and suggestions for improvement. 

1. I am still concerned about the primary endpoint – will any lower respiratory tract specimen be used? There are data suggesting differences between BAL, endotracheal tube aspirates, and sputum. What is a patient is no longer intubated at 72 hours? Will their BAL/endotracheal aspirate be compared to a sputum sample at 72 hrs? Sample type appears to be a confounder for interpretation of the results. Additionally, in line 159 the endpoint is still listed as endotracheal aspirate.

We have not attempted to limit lower respiratory tract specimens, since some sites may prefer a BAL, others a mini BAL, etc. The key point will be that the same approach will be used in both arms, such that this will not be a confounder.

Because the typical duration of mechanical ventilation for this patient population is between 8 to 10 days, we do not expect a large number of patients to be extubated before 72 hours. Nevertheless, if there are unexpected quicker recoveries, we would expect this to balance out between the two groups, barring a miraculous effect from the experimental treatment.

We have corrected the terminology in line 159.

2. There is no discussion of how administration of antiviral therapy may affect the virologic endpoint. The population of interest (mechanically ventilated adults) would not uniformly receive remdesivir given current guidelines, however, some may either before study enrollment or after. Stratification could be an option.

Because the treatment of this patient population is undergoing constant change, with different approaches emerging, and previous treatments being found to have essentially no effect (for example, remdesivir), we have avoided making explicit requirements for treatment. We instead rely on the fact that over the duration of the study, the specific treatments provided (and the changes in treatments that occur as new data or treatments emerge), will occur essentially equivalently between the two randomized groups. If there is a substantial imbalance, additional stratification will then likely be necessary. We expect that this pilot will give valuable baseline information in this regard.

---

## [Decision Letter · Decision Letter 2]

18 Nov 2020

Core warming of coronavirus disease 2019 (COVID-19) patients undergoing mechanical ventilation: protocol for a randomized controlled pilot study

PONE-D-20-12959R2

Dear Dr. Kulstad,

We’re pleased to inform you that your manuscript has been judged scientifically suitable for publication and will be formally accepted for publication once it meets all outstanding technical requirements.

Kind regards,

Steven Eric Wolf, MD

Academic Editor

PLOS ONE

Additional Editor Comments (optional):

Reviewers' comments:

Reviewer's Responses to Questions

**Comments to the Author**

1. Does the manuscript provide a valid rationale for the proposed study, with clearly identified and justified research questions?

Reviewer #1: Yes

Reviewer #2: Yes

2. Is the protocol technically sound and planned in a manner that will lead to a meaningful outcome and allow testing the stated hypotheses?

Reviewer #1: Yes

Reviewer #2: Yes

3. Is the methodology feasible and described in sufficient detail to allow the work to be replicable?

Reviewer #1: Yes

Reviewer #2: Yes

4. Have the authors described where all data underlying the findings will be made available when the study is complete?

Reviewer #1: Yes

Reviewer #2: Yes

5. Is the manuscript presented in an intelligible fashion and written in standard English?

Reviewer #1: Yes

Reviewer #2: Yes

6. Review Comments to the Author

You may also provide optional suggestions and comments to authors that they might find helpful in planning their study.

Reviewer #1: No further comments to add as I had said when I looked at the 1st revision. My understanding of this work is to simply check what warming of severely sick COVID-19 patients may do in a small feasibility/pilot study.

Reviewer #2: The authors have adequately addressed all of my comments and I have no further concerns for publication.

7. PLOS authors have the option to publish the peer review history of their article (what does this mean?). If published, this will include your full peer review and any attached files.

Reviewer #1: **Yes: **Greg Fegan

Reviewer #2: No

---

## [Editor Report · Acceptance letter]

20 Nov 2020

PONE-D-20-12959R2 

Core warming of coronavirus disease 2019 (COVID-19) patients undergoing mechanical ventilation – a protocol for a randomized controlled pilot study

Dear Dr. Kulstad:

I'm pleased to inform you that your manuscript has been deemed suitable for publication in PLOS ONE. Congratulations! Your manuscript is now with our production department. 

Kind regards, 

on behalf of

Dr. Steven Eric Wolf 

Academic Editor

PLOS ONE